

# Morphology and Raman spectra of aerodynamically-classified soot samples

Alberto Baldelli, Steven Nicholas Rogak[1]

[1]Department of Mechanical Engineering, University of British Columbia
6250 Applied Science Ln #2054, Vancouver, BC V6T 1Z4

*Correspondence to*: Alberto Baldelli (baldelli.alberto@yahoo.com)

**Abstract.** Airborne soot is emitted from combustion processes as aggregates of "primary" particles. The size of the primary particles and the overall aggregate size control soot transport properties, and prior research shows that these parameters may be related to the soot nanostructure. In this work, a laminar, inverted non-premixed burner has been used as a source of soot that is almost completely elemental carbon. The inverted burner was connected to an Electrostatic Low-Pressure Impactor, which collected particles on stages according to the aerodynamic diameter, from 0.3 to 10 µm. The morphology was analyzed using a Transmission Electron Microscope followed by image processing to extract projected area and average primary particle size for each aggregate (approximately 1000 aggregates analyzed in total for the 9 impactor stages). Carbon nanostructure was analyzed using a Raman spectrometer, and 5 absorption bands (D4, D1, D3, G, and D2) were fitted to the spectra to obtain an estimate of the carbon disorder. The average primary particle diameter increases from 15 to 30 nm as the impactor stage aerodynamic diameter increases. The D1, D3, D2, and D4 bands decreased (relative to the G band) with the particle size, suggesting that the larger aggregates have larger graphitic domains.

## 1    Introduction

Many studies on carbonaceous aerosols, their production, properties, and impacts on the environment and human health have been conducted (Pöschl, 2005). Considering the complexity of the soot formation processes, the diversity of sources that produce it, and the variety of measurement approaches used, it is not surprising that the literature contains a wide range of measured values for soot size, optical properties, and chemical reactivity (relevant to toxicity, behavior in the environment, and behavior in engineered systems such as oxidation traps).

Soot is produced through the rapid pyrolysis of fuel molecules followed by nucleation, surface growth, aggregation and oxidation (Frenklach, 2002). Typically, mature soot is composed of graphene layers held together by weak van der Waals interactions, separated by a distance of approximately 0.335 nm (Frenklach, 2002). However, in addition to this structure, soot can contain both amorphous carbon and crystalline fullerenic carbon (Vander Wal et al., 2010). Amorphous soot consists of short, disconnected, and randomly oriented graphene segments (Jaramillo et al., 2014). The amorphous soot includes PAHs and other components, such as aliphatics, sulfates, and metal oxides. The ratio between the amorphous organic carbon and the




crystalline graphite-like carbon relates to the condition of the soot formation process (Sadezky et al., 2005). The processes described here lead to the formation of nearly spherical "primary particles" of 10-40 nm diameter that join together through Brownian coagulation into aggregates 50-1000 nm long. Apparently, the majority of this coagulation happens over very short length and timescales, because within each aggregate the primary particle size is quite uniform. This is suggestive of uniform

conditions present for the formation of each aggregate (Dastanpour and Rogak, 2014), which would imply that the primary particle size might be correlated with soot chemical and optical properties. If this is true, then variations in soot aggregate properties could provide information on variations in flame reaction zones. A correlation between particle size and chemical/optical properties would have implications for the way that soot is measured and how it would behave in the environment. The present work takes a step towards understanding these correlations by applying Transmission Electron

Microscopy (TEM) and Raman spectroscopy to soot segregated by aerodynamic diameter.

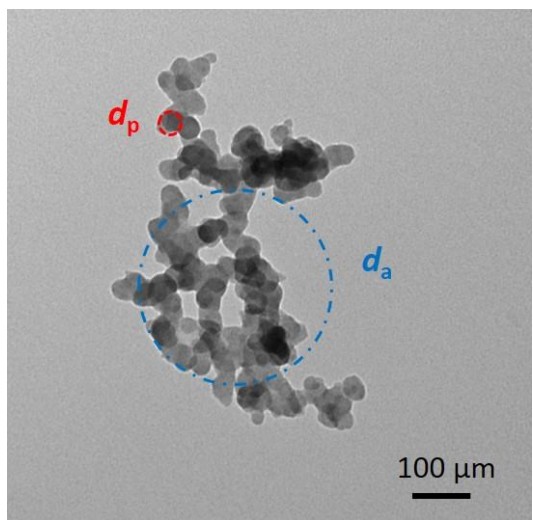

Figure 1 Illustration of the terms primary particle diameter ($d_p$, red dashed circle) and projected-area-equivalent diameter ($d_a$, blue dot dashed circle). The image is from stage 6 of the ELPI for an ethylene flow of 0.13 lpm.

Laboratory burners are commonly used for studies on the properties of soot. Premixed flame burners, such as the McKenna burner, require globally rich mixtures to produce soot; non-premixed flame burners, such as the Santoro burner or inverted burner (Stipe et al., 2005) are a better model of practical combustion devices in that they produce soot with globally lean

mixtures. Inverted burners are advantageous because they produce very steady flames with high soot yields (Ghazi et al., 2013). In this work, we applied the TEM image analysis and Raman spectroscopy to study soot produced by an inverted burner and subsequently segregated by size..



## 2    Materials and Methods

### 2.1    Soot nanoparticles production and sampling

Soot was produced with a miniature inverted burner (Argonaut Scientific) using ethylene (0.13 lpm) burning in bottled air (10 lpm). The inverted burner is sealed during operation to avoid flickering flames. A tee connection at the exhaust of the burner

ensures that the burner operates at atmospheric pressure (Figure 2). A pump is used to supply a dilution line to reduce concentrations at the instruments.

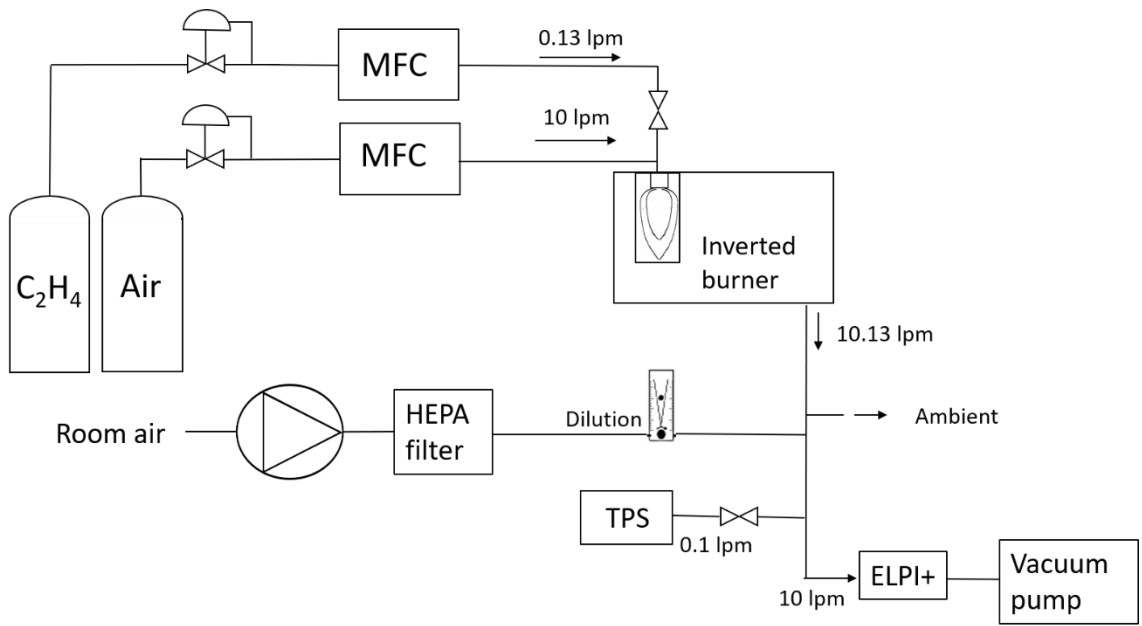

Figure 2 Schematic of the setup used for the production and sampling of soot. MFC and TPS refer to Mass Flow Controller and Thermal Particle Sampler, respectively.

An Electrical Low Pressure Impactor (Dekati ELPI+) was used to collect size-segregated samples, as in some earlier work (Kim et al., 2013; Liati et al., 2018). The ELPI+ also provides electrical currents from each stage that are used to recover aerosol size distributions. For determining size distributions from the currents, an acquisition time of 120 seconds was used,

with a dilution flow of 2 lpm. For collection of samples for TEM (which required a light loading), a sample time of 5 seconds was used with a dilution flow of 3 lpm. For the Raman samples on titanium foils, a sample time of 10 seconds was used with a dilution flow of 3 lpm. TEM and Raman samples were collected from stages 3 to 11 of the ELPI+. Raman spectra of soot aerosols were collected at least 30 minutes after burner ignition to ensure flame stability. A Carbon Type-B- 300 mesh Copper,

1813, Ted Pella has been used as a collection substrate for the TEM analysis, while a titanium foil of $3 \times 3$ mm for the Raman analysis. Both types of substrates were placed on each stage using a piece of tape applied only at the edges of the grid. Impactor



grease was not applied in the area of the collection substrates. The impactor grease reduces the aggregates bouncing from one stage to the following one.

A Thermophoretic Particle Sampler (TPS) was used to collect unsegregated (total) samples from the exhaust of the burner. For samples collected by the TPS and analyzed by Raman microscopy, a sample time of 30 seconds was used with a dilution flow of 4 lpm. This sampler heats the aerosol to approximately 200ºC in a small capillary (residence time of <0.1 s) before the sample is deposited on a room-temperature microscope grid (or titanium foil).

## 2.2 Sample analysis

TEM has been widely used to analyze the morphology of soot nanoparticles. Manual or automatic codes are commonly used to pursue a detailed analysis of TEM images (Wang et al., 2016; Yehliu et al., 2011). Manual sizing is laborious which makes it difficult to generate statistically adequate measurements. These issues can be solved by using an automatic codes, but usually with a loss of accuracy in the measurements for each aggregate (Wang et al., 2016).

A Hitachi H7600 TEM was used to produce at least 50 images per ELPI+ stage. In this work we use the algorithm developed by Dastanpour et al. (Dastanpour, 2016). The code is used in the current work to determine the projected area-equivalent diameter of the aggregates and the average primary particle diameter for each aggregate (Figure 1 shows an example of a primary particle highlighted in red within a complete aggregate). The projected area-equivalent diameter is the diameter of a sphere with the same projected area as the aggregate. The algorithm determines the average primary particle diameter for each aggregate with an error of less than 14% relative to manual sizing, for individual aggregates, and a bias of less than 4% for typical samples.

TEM images are analyzed using two different procedures. TEM images can be defined "overloaded", as shown in Figure 3 a, which shows soot aerosols collected on the ELPI stage 6 and generated by an ethylene flow of 0.13 lpm using the system shown in Figure 1. Red dot dashed ovals identify overloaded areas where it is impossible to use our image analysis codes. Blue dashed circles identify the areas selected for imaging. Imaging only soot aggregates located in the blue dashed areas may exclude large aggregates that tend to follow the impactor streamlines and to impact onto other large aggregates creating larger clusters. Thus, we expect these TEM samples to bias the results to smaller sizes than the true mass-averaged size.




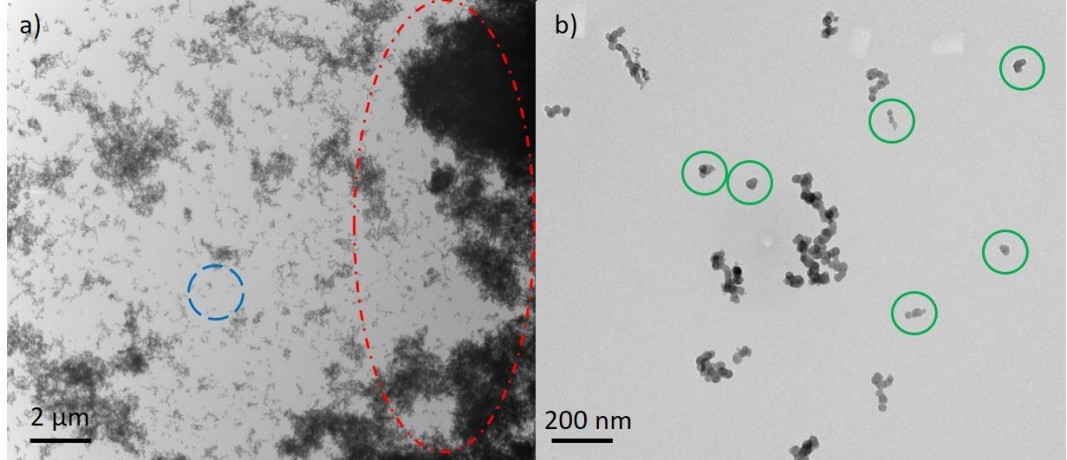

Figure 3 TEM images of aerosols collected on stage 6 for a) low dilution, and b) more diluted sample[1]. For both cases the ethylene flow was 0.13 lpm. In a) red dot dashed circles identify overloaded areas, and blue dashed circles identify the area selected for image processing. In b) green circles identify aggregates composed by less than three soot nanoparticles that are removed by the image analysis.

Decreasing the collection time and increasing the dilution rate improved the sampling considerably producing "low-loaded" samples. Figure 3 b) (for stage 6) has been produced using a dilution rate of 2 lpm, a purge flow of 2 lpm, and a collection time of 2 seconds; for those tests the bottled air was replaced with building air, shown in the SI (Figure S.3). Using this

alternative setup system, TEM grids have been placed on stage 4, 6, 8, and 10. TEM images show much less populated samples, Figure 3 b). However, now we noticed the presence of some aggregates much smaller than expected (green circles). For stage 6, the expected mean aerodynamic diameter would be 120 nm; the aggregates contained in the green circles in Figure 3 b) are approximately 20 nm. A recent work shows the impaction effect of micro and nanoparticles on impactor stages; large aggregates seem to break into smaller aggregates while impacting onto the stage (Wernet et al., 2017). Conceivably a similar

mechanism could apply to the ELPI+ sampling. These small aggregates, although surprising, have little influence on the results for either the TEM (because primary particle sizes are plotted as a function of aggregate size) or Raman analysis (which we expect to reflect mass-averaged characteristics). Given the potential of these small particles being artifacts, we have removed them in the reported averages for the stages. Specifically, aggregates lower than the 15% of the average equivalent area diameter at each stage and at each operating condition of the inverted burner. The 15% is an arbitrary cutoff but small

differences are encountered if values between 10 to 20% are selected.

Raman spectroscopy has been used to find the ratio between the amorphous and the crystalline contents of soot nanoparticles (Sadezky et al., 2005). Raman spectroscopy is sensitive not only to crystal structures but also to molecular structures (short-

---

[1] The diluted samples were created using filtered building compressed air for combustion rather than the bottled air. There were indications that this may have affected Raman spectra, but the overall conclusion (see supplemental information) was that the air source has negligible effect on the soot.





range order) (Sadezky et al., 2005). Here, we assume that Raman spectra can be taken as an indication of nano-structural features relevant to optical or chemical properties. The Raman spectrum of soot shows clear bands, each one with a specific reference to the chemical properties of soot (Saffaripour et al., 2017). By knowing the fingerprints of soot nanoparticles may be possible to recognize the influence of the production conditions, as gas type, flame location, flame velocity, and source type

(Patel et al., 2012).  The titanium substrate was selected for the Raman analysis since it is Raman inactive and oxidation is much slower than for most metals. The titanium foils can be punched to produce disks of 3 mm, which can be substituted for common TEM grids in the TPS.

The Raman spectrometer used was a Renishaw Confocal with a digital stage and 785 nm point-focus laser at typical power of

0.2 mW. Three or more Raman spectra were collected for each sample in order to generate a standard deviation (later, the error bars presented on the graphs are from error propagation of these standard deviations). The exposure time was 10 seconds, and the accumulation time was 1 second. The soot Raman spectra were fitted using Origin Pro software as follow. Spectra were smoothed over the 500 to 2000 cm$^{-1}$ range selected for fitting of the first-order soot Raman bands. The roughness of the titanium disks and their oxide layer generate a Raman spectrum background, which  was used as a baseline in order to eliminate

the spectra slopes (Calabrese et al., 2017). An approximate location for the peak centers was selected, but a change of the peak centers was allowed within the expected ranges shown below. Blank samples with and without the impactor grease showed a near-linear trend that was removed by the baseline subtraction process described above. The five-peak deconvolution has been used and validated in previous literature references (Lapuerta et al., 2012; Lapuerta et al., 2011; Sadezky et al., 2005; Saffaripour et al., 2017; Seong and Boehman, 2013). The five main peaks in soot (Lapuerta et al., 2012; Lapuerta et al., 2011;

Sadezky et al., 2005; Saffaripour et al., 2017) are as follows:

> *D4* (centered 1127 cm$^{-1}$ to 1208 cm$^{-1}$) - Caused by curved PAHs layers in graphitic crystallites, carbon atoms in sp$^3$ and intermediate sp$^2$ – sp$^3$ hybridization states;

> *D1* (centered 1301 cm$^{-1}$ to 1317 cm$^{-1}$ for a laser wavelength of 780 nm) - Induced by defects along graphitic edge planes;

> *D3* (centered around 1489 cm$^{-1}$ to 1545 cm$^{-1}$) - Induced from impurity ions (ex. calcium, fluorine, and potassium), and
amorphous carbon mixtures;

> *G* (centered around 1571 cm$^{-1}$ to 1598 cm$^{-1}$) - Identifies the graphitic content in soot nanoparticles;

> *D2* (centered around 1610 cm$^{-1}$ to 1625 cm$^{-1}$) – Related to the disorder in polycyclic aromatic hydrocarbons (PAHs) around the soot boundaries.

The combination of a Gaussian curve for D3 band and Lorentzian curve for D4, D1, D2 and G bands, all with floating peak locations, was found here to generate the lowest chi-squared sums, the highest linear regression coefficients, and very repeatable results from sample to sample. The number of free parameters used in these fits is comparable to those used by (Saffaripour et al., 2017) and others that used five bands. Figure 4 shows a typical curve fitting of a Raman spectrum of soot



nanoparticles deposited on a stage (stage 4 is used as example). The Supplemental Information Table S.1, S.2, and S.3 provides the details of the fits and peaks for the samples.

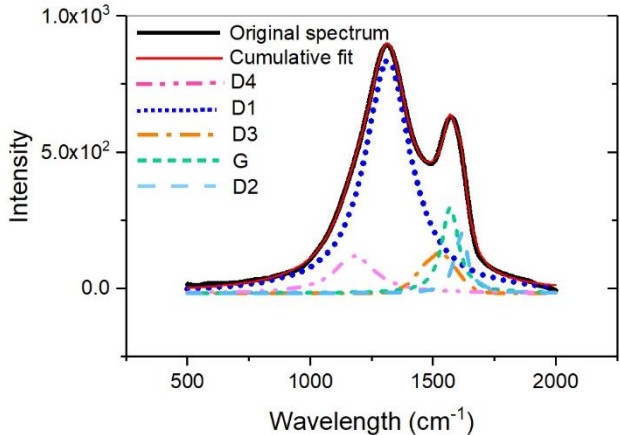

Figure 4 Five bands curve fitting of a typical soot Raman spectrum. D1, D2, D4, and G bands are fitted with a Lorentzian curve, while D3 band is fitted with a Gaussian curve. The sample is from soot nanoparticles generated by an inverted burner and collected with an ELPI+ (Stage 4) with the system shown in Figure 1.

## 3    Results and discussion

Soot aerosols were characterized by their aerodynamic size distribution, electron microscopy of samples, and Raman spectroscopy. The results discussed below were obtained using the conditions explained in Section 2.1 and the apparatus shown in Figure 2 and, only for TEM samples, an alternative system shown in Figure S.4 in the SI including building air instead of bottled air.

### 3.1    Aerodynamic size distribution

Figure 5 shows the number distribution as a function of aerodynamic diameter. In Figure 5, the mean aerodynamic diameter $D_{ae}$ (in microns) are 0.02, 0.04, 0.07, 0.12, 0.2, 0.31, 0.48, 0.76, and 1.25 for the Stages 3 to 11 respectively. The particle density used in the ELPI+ data inversion was set to 1.2 g/cc based on work in (Maricq and Xu, 2004). In fact, the effective density of soot decreases with size, but we still expect that most of the *mass* is on largest stages. At the burner condition used, the flame tip is "open", soot emissions are very high, and super aggregates can form (Ghazi et al., 2013; Saffaripour et al., 2017). Figure 5 shows the distribution of the aerodynamic diameter at each stage of ELPI+. The aerodynamic diameter distribution shows a peak at Stages 5 to 7; most aggregates produced by the inverted burner are between 0.1 to about 5 µm. Stages 8, 9, and 10 shows the highest mass content, as shown in the supplementary material by observing Figure S.1.





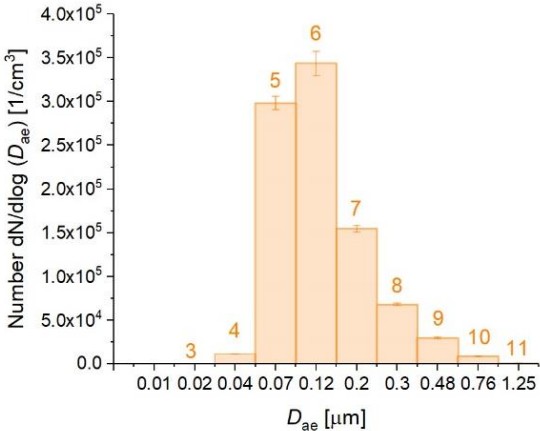

Figure 5 Number distribution recorded by ELPI+ for stages 3 to 11, with the stage mean aerodynamic diameter given below each column.

## 3.2    TEM analysis

In Figure 6, the relationship between the projected-area-equivalent diameter ($d_a$) and the primary particle diameter ($d_p$) is shown. For comparison, we show the "universal fit" (Olfert and Rogak, 2019) for a wide range of soot sources; the fit equation is

$$d_p = 17.8 \left(\frac{d_a}{100}\right)^{0.35}$$
Equation 1

Results from the samples collected here from the ELPI+ are quite close to Equation 1 (dashed line), but depending on which parts of the impactor deposits are analyzed, there are some discrepancies. In Figure 6, full symbols identify the method and system explained in Figure 3.a, empty symbols in Figure 3 b. In the case of full symbols, the TEM grids were often overloaded, such that imaging needed to be done in areas away from the main deposits on the impactor stages, Figure 3 a. Thus, there could be bias in the choice of particles imaged, but in fact the TEM sizing is reasonably consistent with the impactor stage. For

example, the mean area-equivalent diameter for stage 5 is 140 nm. Aggregates of this size should have an effective density of approximately 430 kg/m$^3$ (using the Olfert and Rogak relations), resulting in an aerodynamic behavior equivalent to an 83 nm sphere with density of 1200 kg/m$^3$ (using projected-area diameter as a proxy for mobility diameter). Given the uncertainties involve in the effective density and the resolution of the ELPI (and TEM), this appears consistent with the nominal midpoint aerodynamic diameter for this stage: 70 nm. For stage 10, the area-equivalent diameter is 208 nm, which we expect to result

in an aerodynamic diameter of 112 nm. This far lower than the stage midpoint of 760 nm – most likely an indication that the TEM imaging is biased towards unusually small aggregates that deposit on the stage away from the stagnation point by a mixture of inertial effects and diffusion.  This interpretation is supported by additional sampling for similar burner operation, but much lower concentrations, as shown in the empty symbols in Figure 6. In this case, TEM measurements followed Equation 1 quite closely up to aggregate sizes of a micron –closer to but still smaller than the expected impactor stage cut point.  The

comparison between the heavily-loaded and lightly-loaded cases clearly shows that the sizing can be biased, but in either case





the primary particle size increases monotonically with the stage number, so differences in Raman spectra with the stage number are plausibly associated with primary particle size.

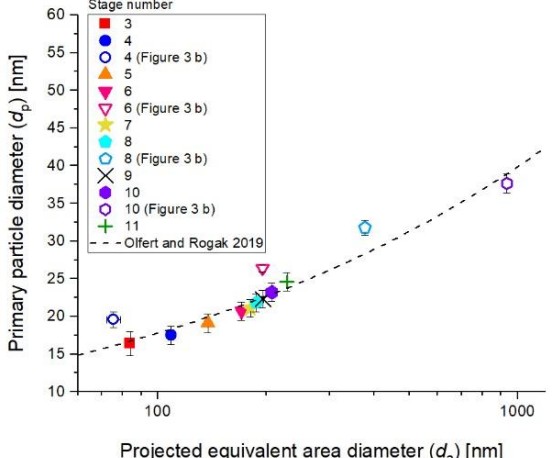

Figure 6 The relationship between the primary particle diameter and the projected-area-equivalent diameter from the literature is shown as the dashed line (Olfert and Rogak, 2019) and experimentally here (symbols). Full and empty symbols identify results achieved by analyzing similar images as Figure 3 a) and b), respectively.

### 3.3 Raman spectroscopy

5 Figure 7 shows the Raman spectra (normalized by the D1 peak) of soot aerosols collected on different stages and from a sample of unclassified soot collected by a thermophoretic sampler (TPS) onto the same type of titanium foil ("Total"). The spectra are very similar, and it is only when ratios are plotted that differences become apparent.



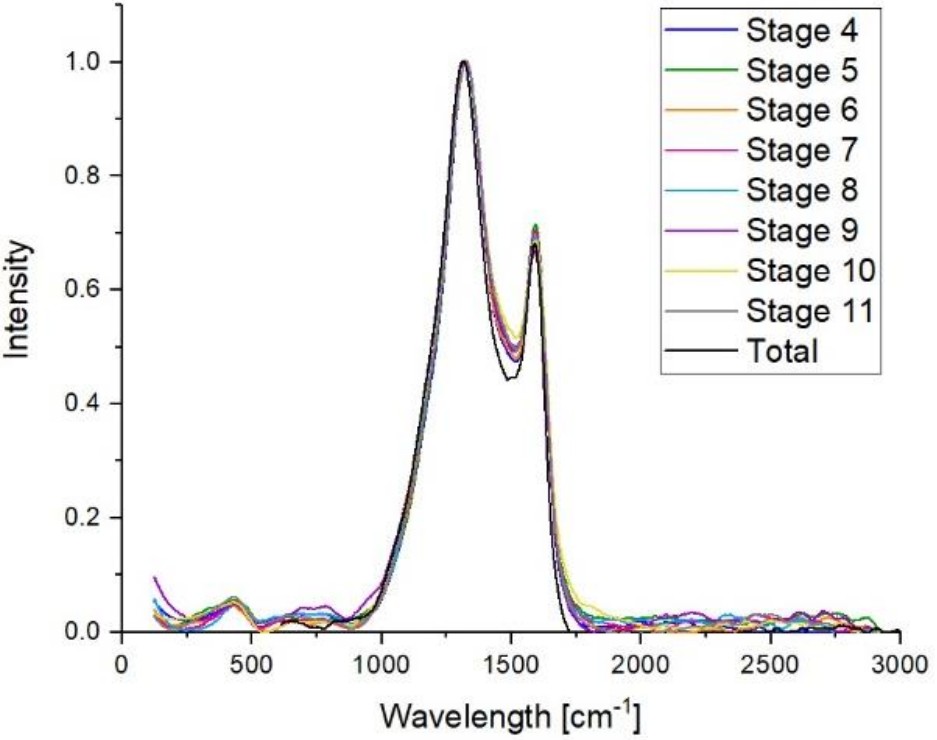

Figure 7 Normalized Raman peak ratios of soot segregated by a sampling on the stages of the ELPI+ impactor. Raman spectrum of the total exhaust ("total") is derived from samples collected by the Thermophoretic Particle Sampler (TPS).

Figure 8 shows D1/G, D3/G, D2/G, and D4/G peak ratios based on the areas of the fitted bands; these are generally consistent with the values reported in previous literature references (Patel et al., 2012; Sadezky et al., 2005; Saffaripour et al., 2017). Figure 8 suggests that the D3/G and D1/G band ratios decrease with particle size. Other peak ratios, as D2/G and D4/G

moderately decrease. A straightforward interpretation of this is that the amorphous carbon content of the soot is smaller for larger particles. This confirms the expectation from (Dastanpour, 2016) that the larger particles would be more graphitic. Samples were collected for other conditions but are not included here because the burner warmup time was not consistent. However, those samples also show, qualitatively, a similar trend between the Raman band ratios and the particle size (Supplementary Information, Figure S.5). Furthermore, one series of experiments was conducted using compressed building

air for the burner supply. Samples from these experiments did NOT show the same trends with stage number (Supplemental Information Tables S.4 and S.5), but when the aerosol was run through a thermodenuder the expected falling trend of D1/G with stage was recovered (Tables S.6 and S.7). Apparently, contamination in the supply air produced a coating on the particles that affected the Raman spectra. In all cases, the ratios for the unsegregated samples (from TPS, labeled "total") are close to the average for all stages – as expected.





The ratios shown in Figure 8 are derived from peak fitting to the average of 3 Raman spectra. The smoothed average of these 3 spectra are fitted to the 5 bands in Origin Pro, using the standard deviation of the triplicates as the weighting in the Levenberg-Marquardt (L-M) algorithm. The error bars for the ratios (typically lower than 20%) in Figure 7 are derived from error propagation of the fit uncertainties in the peak areas reported by Origin Pro. These fit uncertainties are the standard errors

produced through the L-M algorithm. Actually, the repeatability of the procedure is better than indicated by the error bars. For example, when 4 samples were re-analyzed after about 5 months and re-fitted, D1/G ratio increase by a mean of 2.6 %.

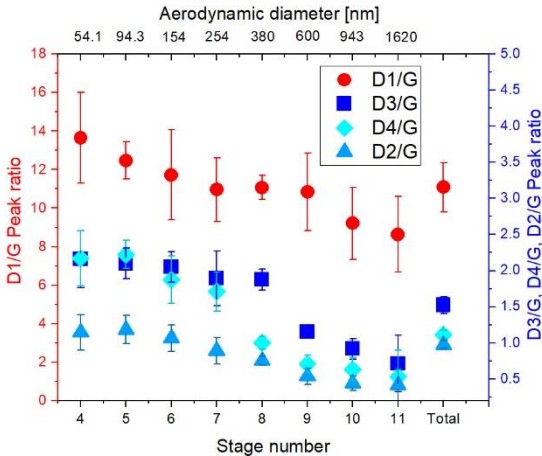

Figure 8  Raman peaks ratios of soot segregated by an ELPI+ impactor are shown. Raman peaks ratios of soot produced for ethylene flow 0.13 lpm collected on titanium grids on the stages 3 to 11. The Raman peak ratios of the total exhaust ("total") are reported as well.

## 4   Conclusions

Raman spectroscopy and TEM analysis were performed on samples of soot classified by aerodynamic diameter. The primary

particle diameter increases for larger projected area diameters; the relationship is in approximate agreement with a correlation developed recently for multiple soot sources (Olfert and Rogak, 2019). The challenge in validating this relationship quantitatively using the impactor sampling is that there are potential artifacts due to overloaded samples and possible fragmentation.  However, it is clear that the upper stages (larger aerodynamic diameter) contained aggregates with much larger primary particles.

Raman spectra showed a consistent variation from the bottom (small aerodynamic diameter) to top (large aerodynamic diameter) stages. The spectra for the total (unclassified) soot samples are similar to the spectra for the middle stages, which makes sense given that the middle stages collect most of the aerosol mass. Fitting the Raman spectra to a 5-band model results in decreasing D1/G, D2/G, D3/G and D4/G ratios with size. This would imply that the larger particles contain larger graphitic



domains and are more ordered. The most plausible explanation for this shift in the Raman spectra is the associated shift in primary particle size for each stage, rather than the changing aggregate size *per se.*

This work provides evidence that both soot primary particle size and nanostructure depend on the aggregate size. This suggests

that samples collected post-flame still carry information about local formation conditions in the flame. It also means that conventional approaches to modelling fractal aggregates are incorrect in assuming that size and soot material properties are independent.

## 5    Supplemental material link

Supplemental material includes additional results on the soot sampling and the repeatability of the experiments. The

supplemental material is uploaded in the following link.

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
