# Peer review of "Morphology and Raman spectra of aerodynamically-classified soot samples"

_Atmospheric Measurement Techniques, 2019_

## Referee Comment (RC1) · Anonymous Referee #1 · 3 Jun 2019

The authors study of the properties of size- filtered soot particles performed by means of electron microscopy and Raman spectroscopy. The authors are able to show that the overall aerodynamic size of the soot aggregates is correlated to the size of the spherical primary particles that are the building blocks of the aggregates. Raman spectroscopy reveals that larger soot aggregates exhibit more crystalline graphitic domains as compared to smaller aggregates. The authors convincingly attribute this finding to the larger primary particle size found in the larger aggregates. This is probably the most important result of the study. The study usesand appropriate technology and is carefully performed and analyzed according to the established methodological standards. Recommendation: As stated by the authors, many similar studies have been

performed earlier. It is important to compile the results of comparable studies more extensively and to clarify explicitly, where the study goes beyond previous work. Apart from that, the manuscript is comprehensive and well written and I do not have further comments.

---

## Author Comment (AC1) · 14 Jun 2019

We would like to thank the reviewer for the interest in our publication. We carefully considered the reviewer's comment and we agree with it. Even though we mentioned that some previous studies that encountered a correlation between the morphological and chemical properties of soot aggregate, we agree that these literature references should be cited. Therefore, the following sentence was added at page 2 and line 9: "This correlation has been previously observed in few literature references (Saffaripour et al., 2017, Dastanpour and Rogak, 2014, Alfe' et al. 2009, Mühlbauer et al. 2016). The present work, however, takes a step towards understanding these correlations by

applying Transmission Electron Microscopy (TEM) and Raman spectroscopy to soot segregated by aerodynamic diameter."

---

## Referee Comment (RC2) · Anonymous Referee #2 · 17 Jun 2019

This is an interesting and well-written paper focusing on the relationship between Raman spectroscopy and morphology and structure of soot particles. The authors use TEM and ELPI+ in order to gather the particle morphology. Raman spectroscopy has been used to determine structural information. The subject of the manuscript fits well into AMT, but conclusions are rather short and somewhat weak. I recommend publication after major revisions of the conclusion section.

Specific Comments

Abstract, p1, L15: "5 absorption bands" this is simply wrong. Raman in an inelastic scattering technique and is not related to any absorption phenomena. Just say "5

vibrational bands"

P1, L29: "crystalline fullerenic carbon" is not the right wording. Soot either includes crystalline graphite-like carbon, amorphous carbon and very rarely fullerenes. Incomplete fullerene structures (fullerenic carbon?) are caused by defects and are non-planar (incomplete sp2 hybridized) and therefore are amorphous.

P2, L7: "A correlation between particle size….." You might quote here a recently accepted AMTD paper: Haller, T., Rentenberger, C., Meyer, J. C., Felgitsch, L., Grothe, H., and Hitzenberger, R.: Structural changes of CAST soot during a thermal-optical measurement protocol, Atmos. Meas. Tech. Discuss., https://doi.org/10.5194/amt-2019-10, in review, 2019.

P2, L12: "Laboratory burners…." You might discuss similarities and differences regarding the CAST burner, which is the most commonly used burner for laboratory soot studies. P5, L18: "Raman spectroscopy is sensitive not only to…" better write "Raman spectroscopy is sensitive only to short-range order, molecular structures but due to the symmetry of the observed vibrations also structures and morphologies can be differentiated (Sadezky et al. 2005).

P6, L5: "The titanium substrate was selected…" better write "…since titanium and TiO2 exhibit no Raman active vibrations in the area of interest…"

P6, L14: Explain how you subtracted the fluorescence of the soot.

General Comments

Don't use the word "peak ratios" when describing "band ratios". Other phrases are: intensity ratios, ratios of band areas, etc.

Transfer "lpm" into "sccm"

---

## Author Comment (AC2) · 20 Jun 2019

We appreciate the detailed comments of the reviewer and we carefully considered each of the comments suggested. Abstract, p1, L15: "5 absorption bands" this is simply wrong. Raman in an inelastic scattering technique and is not related to any absorption phenomena. Just say "5 vibrational bands" As pointed out by the reviewer, the term "absorption" was wrongly used when associate it with Raman spectroscopy. Therefore, the term absorption was substituted it with the suggested term "vibrational"

P1, L29: "crystalline fullerenic carbon" is not the right wording. Soot either includes crystalline graphite-like carbon, amorphous carbon and very rarely fullerenes. Incom-

plete fullerene structures (fullerenic carbon?) are caused by defects and are non-planar (incomplete sp2 hybridized) and therefore are amorphous. As in the previous comment, a wrong wording was used. As suggested by the reviewer, the wording "crystalline fullerenic carbon" is substituted with "crystalline graphite-like carbon".

P2, L7: "A correlation between particle size. . ..." You might quote here a recently accepted AMTD paper: Haller, T., Rentenberger, C., Meyer, J. C., Fel-gitsch, L., Grothe, H., and Hitzenberger, R.: Structural changes of CAST soot during a thermal-optical measurement protocol, Atmos. Meas. Tech. Discuss., https://doi.org/10.5194/amt2019-10, in review, 2019. We agree with the reviewer that the suggested reference needs to be added with other previous literature references regarding the correlation between particle size and chemical/optical properties. This literature reference emphasizes the presence of a relationship between particle size and chemical properties of soot nanoparticles. At page 2 and line 10, the following sentence was added: "A few references investigate the chemical and morphological properties soot sampled downstream of a single source operating at constant condi-tions (Alfe' et al. 2009, Haller et al. 2019, Ghazi et al., 2013). However, these studies did not directly correlate the two types of properties, chemical and morphological. The correlation between particle size and chemical/optical properties can be observed by reviewing the results of a few literature references (Saffaripour et al., 2017, Ess et al. 2016). The present work takes a step towards confirming, verifying and understanding these correlations by applying Transmission Electron Microscopy (TEM) and Raman spectroscopy to soot segregated by aerodynamic diameter"

P2, L12: "Laboratory burners. . .." You might discuss similarities and differences re-garding the CAST burner, which is the most commonly used burner for laboratory soot studies. The reviewer suggest that CAST burner has to be mentioned while describing alternatives in laboratory burners. Since the CAST burners are extensively used in research studies, we agree that we should discuss the similarities and the differences between CAST burners and inverted burners. Therefore, the following sentence was

added at page 2 and line 17: "Combustion aerosol standard CAST or miniCAST are commonly used since they easy to operate and allow to readily adjust the particle size in a large range, typically between 10 nm and 200 nm (Ess and Vasilatou 2019). Like CAST burners, inverted burners are advantageous because they produce very steady flames with high soot yields (Ghazi et al., 2013). The miniCAST and the mini inverted burner are considered alternative techniques to produce a steady stream of soot particles. The main difference is the lower cost of the mini inverted burner compared to the most popular miniCAST burner."

P5, L18: "Raman spectroscopy is sensitive not only to. . ." better write "Raman spectroscopy is sensitive only to short-range order, molecular structures but due to the symmetry of the observed vibrations also structures and morphologies can be differentiated (Sadezky et al. 2005). As suggested by the reviewer, the sentence at page 5 and at line 18 was modified.

P6, L5: "The titanium substrate was selected. . ." better write ". . .since titanium and TiO2 exhibit no Raman active vibrations in the area of interest. . ." We agree with the reviewer that his or her suggestion would be a better wording. Therefore, the sentence at page 6 and at line 5 was modified as follow: "Since titanium and titanium oxide exhibit no Raman active vibrations in the area of interest, titanium substrates were selected for the Raman analysis".

P6, L14: Explain how you subtracted the fluorescence of the soot. The reviewer is interested in the fluorescence subtraction procedure taken in this publication. We did use a straight line in case residual fluoresce was present. However, due to the characteristics of the soot analyzed, which has a high EC content, the experimental conditions, which include long laser wavelengths, and the subtraction procedure, which involves the use if titanium substrate Raman signal as the baseline subtraction, most of samples analyzed did not show any residual fluorescence. In the main publication, a brief explanation is added in page 6 and line 15 as follow: "By using a long wavelength laser (Grafen et al. 2015) and the titanium substrates Raman signal as baseline subtraction,

most of the samples do not show any residual fluorescence. Otherwise, the residual fluorescence was subtracted using a straight line." General Comments • Don't use the word "peak ratios" when describing "band ratios". Other phrases are: intensity ratios, ratios of band areas, etc. • Transfer "lpm" into "sccm" • Major revisions to the conclusion section As suggested by the reviewer, all the word "peak ratios" were substituted wit "band ratios". However, the unit of lpm was kept since we consider this unit to be more representative and understandable for the broad range of flow rates used both for the combustion fuel gas and for the air. In addition, we agree with the reviewer that the conclusion section might appear weak. As a result, some statements have been emphasized since supported by strong experimental results. However, the length of the conclusion section has not been majorly modified since we would like to leave it concise and direct.
* * *